# Eustress with H_2_O_2_ Facilitates Plant Growth by Improving Tolerance to Salt Stress in Two Wheat Cultivars

**DOI:** 10.3390/plants8090303

**Published:** 2019-08-27

**Authors:** Arafat Abdel Hamed Abdel Latef, Mojtaba Kordrostami, Ali Zakir, Hoida Zaki, Osama Moseilhy Saleh

**Affiliations:** 1Botany and Microbiology Department, Faculty of Science, South Valley University, 83523 Qena, Egypt; 2Biology Department, Turabah University College, Taif University, Turabah Branch, 21955 Taif, Saudi Arabia; 3Department of Plant Biotechnology, Faculty of Agricultural Sciences, University of Guilan, P.O. Box 41635-1314 Rasht, Iran; 4Rice Research Institute of Iran, Agricultural Research, Education and Extension Organization (AREEO), P.O. Box 41996-13475 Rasht, Iran; 5Department of Environmental Sciences, COMSATS University Islamabad, Vehari-Campus, Vehari 61100, Pakistan; 6National Products Department, National Centre for Radiation Research and Technology (NCRRT), Egyptian Atomic Energy Authority, Nasr City, 11787 Cairo, Egypt

**Keywords:** antioxidant enzymes, H_2_O_2_, inter-simple sequence repeat (ISSR), eustress, seawater, wheat

## Abstract

In this study, the positive role of hydrogen peroxide (H_2_O_2_) pretreatment in mitigating the adverse impacts of seawater stress has been evaluated in two wheat (*Triticum aestivum* L.) cultivars, namely Gemmiza 11 as a salt-sensitive and Misr 1 as a salt-tolerant cultivar, with contrasting phenotypes in response to the salinity stress. Under normal conditions, wheat seeds eustress with H_2_O_2_ have shown significant effects on the improvement of plant growth parameters, such as dry weight and root and shoot lengths. Under salt stress conditions, seeds eustress with H_2_O_2_ have shown a reduction in damage to plant growth and physiological parameters as compared to the seeds kept as un-primed in both wheat cultivars. In addition, eustress of seeds with H_2_O_2_ has induced an increment in the pigments content, proline level and mineral uptake (K^+^, Ca^2+^ and Mg^2+^). Moreover, seeds eustress with H_2_O_2_ have shown significant decrement in Na^+^ content uptake in plants and that subsequently reduced lipid peroxidation. Seawater stress has increased the activity of the antioxidant system based on catalase (CAT), peroxidase (POD) and ascorbate peroxidase (APX) in both cultivars, except POD in Gemmiza 11. Similarly, the application of H_2_O_2_ has further enhanced the activity of the antioxidant system in stressed plants and this enhancement of the antioxidant system further reduced Na^+^ content in plants and subsequently increased the growth parameters. Results of inter-simple sequence repeat (ISSR) markers have shown clear differentiation among the treatments and have provided strong evidence in support of the hypothesis proposed in this study that H_2_O_2_ eustress improves seed tolerance and enhances plant growth parameters under seawater stress.

## 1. Introduction

Salinity is an environmental factor affecting about one-third of the agricultural lands in the world and is considered as a serious problem for crop production in arid and semiarid regions [1,2,3]. In these regions, the water shortage, limited rainfall, intense heat, high evapotranspiration, poor water quality, improper agricultural practices and unmanaged irrigation systems have more seriously raised this problem [4]. On the other hand, the widespread use of irrigation leads to the penetration of seawater into irrigated water; and thus, freshwater becomes increasingly saline [5]. In order to cope with the lack of freshwater for sustainable agricultural development, agricultural scientists and planners knowledge of the utilization of seawater, at least diluted, is essential for the proper irrigation of agricultural crops [6,7].

Salt stress-induced shortage water can cause oxidative stress by increasing the production of reactive oxygen species (ROS), which results in cell damage through the oxidation of nucleic acids, lipids and proteins [8]. There is, however, compelling evidence of the biological and signaling role of ROS, especially hydrogen peroxide (H_2_O_2_), as a molecular messenger in plants [9,10]. H_2_O_2_ is one of the non-radical and relatively stable ROS, which is produced in plants during normal aerobic metabolism. At low concentrations, H_2_O_2_ acts as a regulator of some major processes, such as assimilation, photosynthesis, respiration, stomatal conductance, cell cycle, growth and development, and plant response to biotic and abiotic pressures [11]. However, its accumulation above a certain threshold will increase the oxidative damage, and ultimately the cell death [12,13]. Evidences suggest that H_2_O_2_ directly interferes with the expression of many genes and thus causes hypersensitive defense responses [14] or more antioxidant system activity [15] in plants under environmental stress conditions. Under salinity stress conditions, nitric oxide and H_2_O_2_ as messenger molecules cause the ionic balance in plant cells and cause salinity stress resistance. These molecules adjust the K^+^/Na^+^ ratio in plants, thereby avoiding plant stress from salinity [16]. In general, H_2_O_2_ accumulates in plants when the stress occurs, and a number of reports have suggested that H_2_O_2_ is a key factor in the phenomena of assimilation and stress adaptation [17]. Pretreatment of the seeds can help in controlling the process of water loss, and enhance metabolic activities before free radicals accumulate [18].

Molecular markers have been used to study the natural diversity of species, plant ecotypes and cultivars or the variations caused by induced mutations in plants [19,20]. Amplification of inter-simple sequence repeat (ISSR) molecular markers does not require prior knowledge of the genome and the design of specific primers [21], as microsatellite core sequences can be used as primers in the ISSR-PCR. However, unlike microsatellite markers, the knowledge of a host genome is not essential for the design of ISSR markers [19,20,22]. ISSR technique is dominant, more stable and reproducible, so this technique is used for many purposes such as varietal/line identification, population structure analysis, marker fingerprinting, genetic mapping and phylogenetic assisted selection [23]. The ISSR-PCR can quickly reveal the difference between individuals with a high degree of similarity and include multiple polymorphic loci. ISSR markers have been used widely to detect salt-tolerant genotypes in *Hordeum vulgare* [24], *Sorghum bicolor* [25], *Saccharum officinarum* [26] and *Triticum aestivum* [27].

Wheat (*Triticum aestivum* L.) is one of the most significant crop plants in the world and is relatively salinity-tolerant [28]. In this study, we hypothesize whether wheat seeds eustress with H_2_O_2_ improve tolerance against seawater stress and enhance plant growth parameters. The main aim of this study was to evaluate the impact of seed eustress with H_2_O_2_ on the physio-biochemical responses of two wheat cultivars, Gemmiza 11 and Misr 1, exposed to the seawater stress, which have contrasting salinity-responsive phenotypes. Furthermore, the experiments were performed at the molecular level using six ISSR markers to observe the differential responses of the two wheat cultivars against seawater stress with or without H_2_O_2_-eustress. In fact, it can be said that this is the first report about the application of ISSR markers in determining the effect of H_2_O_2_ eustress on a crop plant at the molecular level. Results from these experiments helped us in understanding the impact of H_2_O_2_ eustress on reducing the adverse effects of seawater stress on tested wheat cultivars as well as the implication of the proposed method in reducing stress in other wheat cultivars and different cereal crops.

## 2. Results

### 2.1. Seed Eustress with H_2_O_2_ Ameliorates Growth and Biomass Production in Wheat Plants

Under non-saline conditions, seedlings eustress with H_2_O_2_ showed high growth rates of both wheat cultivars as compared to the control plants (Figure 1A–C). Seawater stress reduced the seedlings dry weight (DW) (by 0.59 and 0.52 g plant^−1^; Figure 1A), root length (by 8 and 6 cm; Figure 1B) and shoot length (by 4.67 and 5 cm; Figure 1C) of treated plants as compared to the control plants of both cultivars, Gemmiza 11 and Misr 1, respectively. Whereas, eustress with H_2_O_2_ enhanced the seedlings DW (by 0.31 and 0.03 g plant^−1^; Figure 1A), root length (by 4 and 2 cm; Figure 1B) and shoot length (by 2.34 and 1 cm; Figure 1C) in salinized Gemmiza 11 and Misr 1, respectively, in comparison with the seedlings treated with seawater alone.

### 2.2. Seed Eustress with H_2_O_2_ Safeguards Photosynthetic Pigments from Seawater Stress

Pretreatment with H_2_O_2_ boosted the pigment content under the control conditions in both cultivars (Figure 2A–C). In comparison with the control plants, seawater stress reduced chlorophyll (Chl) a (by 0.37 and 0.17 mg g^−1^ fresh weight (FW); Figure 2A), Chl b (by 0.08 and 0.05 mg g^−1^ FW; Figure 2B) and carotenoids (by 0.08 and 0.04 mg g^−1^ FW; Figure 2C) in Gemmiza 11 and Misr 1, respectively. Eustress with H_2_O_2_ enhanced Chl a (by 0.1 and 0.04 mg g^−1^ FW; Figure 2A), Chl b (by 0.03 and 0.2 mg g^−1^ FW; Figure 2B) and carotenoids (by 0.03 and 0.2 mg g^−1^ FW; Figure 2C) in seawater-treated Gemmiza 11 and Misr 1, respectively as compared to seawater plants alone.

### 2.3. Seed Eustress with H_2_O_2_ Modulates the Content of Osmoprotectant (Proline) in Wheat Plants

Eustress with H_2_O_2_ decreased proline content in Gemmiza 11 (by 0.33 mg g^−1^ FW) and increased this osmolyte in Misr 1 (by 0.21 mg g^−1^ FW) (Table 1) as compared to control plants. Seawater stress progressively accumulated proline content in both cultivars (by 1.3 mg g^−1^ FW in Gemmiza 11 and 1.42 mg g^−1^ FW in Misr 1) over the control plants. However, under seawater stress, seed eustress with H_2_O_2_ increased proline content in Misr 1 by 0.17 mg g^−1^ FW and it markedly reduced proline content in the Gemmiza 11 cultivar by 0.55 mg g^−1^ FW relative to that in seawater-stressed-only plants (Table 1).

### 2.4. Seed Eustress with H_2_O_2_ Regulates the Mineral Uptake in Seawater-Exposed Wheat Plants

Under normal conditions, seed eustress with H_2_O_2_ diminished Na^+^ content and increased K^+^, Ca^2+^ and Mg^2+^ contents (Table 1) in both wheat cultivars. Seawater stress markedly and progressively accumulated Na^+^ content in Gemmiza 11 (by 33.72 mg g^−1^ DW) and in Misr 1 (by 18.26 mg g^−1^ DW) as compared to the control plants (Table 1). Interestingly, seawater stress increased K^+^ (by 1.26 mg g^−1^ DW), Ca^2+^ (by 1.83 mg g^−1^ DW) and Mg^2+^ (by 0.57 mg g^−1^ DW) in Misr 1 (Table 1); whereas, seawater stress decreased K^+^ (by 1.1 mg g^−1^ DW), Ca^2+^ (by 0.4 mg g^−1^ DW) and Mg^2+^ (by 0.06 mg g^−1^ DW) in Gemmiza 11 (Table 1) as compared to the control plants. Pretreatment with H_2_O_2_ decreased Na^+^ content in Gemmiza 11 (by 11.86 mg g^−1^ DW) and in Misr 1 (by 3.41 mg g^−1^ DW); on the other side, it enhanced the content of K^+^ (by 6.6 and 1.57 mg g^−1^ DW), Ca^2+^ (by 0.25 and 1.64 mg g^−1^ DW) and Mg^2+^ (by 0.13 and 0.17 mg g^−1^ DW) in Gemmiza 11 and Misr 1, respectively, versus stressed plants (Table 1).

### 2.5. Seed Eustress with H_2_O_2_ Lessens Lipid Peroxidation through the Enhancement of Antioxidant Enzymes Activity in Wheat Plants

H_2_O_2_ eustress reduced the content of malondialdehyde (MDA) (by 4.26 and 9.7 nmol g^−1^ FW) in Gemmiza 11 and Misr 1, respectively, compared to control plants (Figure 3A). Seawater stress induced a significant increase in the content of MDA (by 42 and 10 nmol g^−1^ FW) in Gemmiza 11 and Misr 1, respectively, compared to control plants (Figure 3A). Fascinatingly, H_2_O_2_ application mitigated this increase in MDA content by causing a decrease in Gemmiza 11 (by 28.96 nmol g^−1^ FW) and in Misr 1 (by 9.33 nmol g^−1^ FW) compared with that in the seawater-treated plants alone (Figure 3A). The activity of catalase (CAT), peroxidase (POD) and ascorbate peroxidase (APX) significantly increased in both wheat cultivars eustress with H_2_O_2_ compared to untreated plants (*p* < 0.05; Figure 3B–D). In comparison with the control plants, seawater stress increased the activity of CAT (by 6.62 and 1.1 Unit (U) min^−1^ g^−1^ FW) in Gemmiza 11 and Misr 1, respectively (Figure 3B). H_2_O_2_ application increased the activity of CAT (by 4.47 and 0.9 U min^−1^ g^−1^ FW) in Gemmiza 11 and Misr 1, respectively, versus seawater-treated plants alone (Figure 3B). While seawater stress reduced the activity of POD (by 1.02 U min^−1^ g^−1^ FW) in Gemmiza 11, it increased the activity of POD (by 10 U min^−1^ g^−1^ FW) in Misr 1 in comparison with the water control (Figure 3C). Seed eustress with H_2_O_2_ increased the activity of POD (by 12.79 and 1.33 U min^−1^ g^−1^ FW) in Gemmiza 11 and Misr 1, respectively, compared with that in the seawater-stressed alone plants (Figure 3C). For APX activity, seawater stress increased it (by 1.17 and 3.83 U min^−1^ g^−1^ FW), respectively in Gemmiza 11 and Misr 1 over the control plants (Figure 3D). Eustress with H_2_O_2_ enhanced the activity of APX (by 1.9 and 1.17 U min^−1^ g^−1^ FW), respectively in Gemmiza 11 and Misr 1 in comparison with seawater-stressed-only plants (Figure 3D).

### 2.6. Hierarchical Clustering and Principle Component Analysis (PCA) Analysis

By cutting the dendrogram from 0.06 section, treatments were divided into five groups. Seawater (Gemmiza 11) treatment of 35% was grouped in the first cluster (Figure 4). Based on the studied traits, it was observed that this treatment has the minimum amount for dry weight, root length, shoot length, Chl a and b, carotenoids, K^+^, Ca^2+^, Mg^2+^ and POD. The results showed that salinity stress has a negative effect on these traits in this cultivar. In cluster 2, 35% seawater (Misr 1) and 35% seawater + H_2_O_2_ (Misr 1) treatments were grouped together (Figure 4). Among the treatments, these two treatments had mean values for most of the traits. Although these two treatments were clustered in a group, by observing the traits we can see that in this cultivar, hydrogen peroxide pretreatment is able to significantly reduce the adverse and harmful effects of salinity stress.

In cluster 3, control (Misr 1) and control + H_2_O_2_ (Misr 1) treatments were grouped together (Figure 4). Among the treatments, these two treatments had the maximum values for most of the traits except for MDA and Na content. Although these two treatments were clustered in a group, by studying the traits, we can see that eustress with hydrogen peroxide increased favorable morphological and physio-biochemical traits in this cultivar under the control conditions. In cluster 4, control (Gemmiza 11) and control + H_2_O_2_ (Gemmiza 11) treatments were grouped together (Figure 4). The treatments in this cluster, after cluster 3, had the highest values for most of the traits. Finally, 35% seawater + H_2_O_2_ (Gemmiza 11) treatment was grouped in the last cluster (Figure 4). This cluster, after cluster 1, had the lowest values for most of the traits. Fischer’s discriminant function analysis was used to analyze the accuracy of grouping by cluster analysis. The accuracy of grouping by cluster analysis was 100%.

The PCA was performed to find the association of the different groups of treatments and morpho-physiological and biochemical traits (Figure 5). The two components of PCA (PC1 and PC2) collectively explained 92.37% of the total variation. Results showed five clusters. The first cluster concluded two treatments vis. control + H_2_O_2_ (Gemmiza 11) and control + H_2_O_2_ (Misr 1), respectively. These treatments were associated with the traits including dry weight, root and shoot length and photosynthetic pigments. This means that H_2_O_2_ eustress increased these traits significantly under control conditions. By having an extreme value for most of the traits, these two treatments were selected as the best treatments. Afterward, control (Misr 1) and control (Gemmiza 11) were grouped together. These treatments did not show significant association with any parameters. In addition, 35% seawater (Misr 1) and 35% seawater + H_2_O_2_ (Misr 1) treatments were grouped together. This group associated with antioxidants and proline. Finally, 35% seawater + H_2_O_2_ (Gemmiza 11) and 35% seawater (Gemmiza 11), by associating with Na and MDA, were clustered in two different groups. These two analyses confirmed our results, where seawater stress decreased most of the morpho-physiological and biochemical traits and increased MDA and Na content in two studied cultivars. On the other hand, H_2_O_2_ eustress increased most of the traits under control and salinity conditions.

### 2.7. Molecular Analysis of Treatment

Genetic differences at the DNA level resulted from different treatments which were evaluated using six ISSR primers. Totally, the primers used for diversity analysis of the treatments were able to detect 101 loci, of which 81 (80%) were polymorphic (Figure 6, Table 2, Appendix A). Polymorphic alleles identified by each marker varied from 2 to 23, and an average of 9.1 polymorphic alleles was observed for each ISSR marker. On the other hand, two kinds of changes in the amplified bands were observed where some of the bands disappeared (−) as well as some of the new bands emerged (+) (Appendix A).

The pattern of ISSR-1 primer revealed that under control conditions, seed eustress with H_2_O_2_ treatment led to the production of two bands in Gemmiza 11 and six bands in Misr 1. Under seawater stress, one band in Gemmiza 11 and five bands in Misr 1 were amplified. Under seawater stress conditions, eustress with H_2_O_2_ induced the synthesis of three bands in Gemmiza 11 and eight bands in Misr 1. Eustress with H_2_O_2_ induced the reappearance of one band in salinized Gemmiza 11 and two bands in salinized Misr 1 which disappeared under seawater stress (Appendix A, Figure 6A).

Primer ISSR-2 showed that eustress with H_2_O_2_ caused the synthesis of one band in Gemmiza 11 and two bands in Misr 1 under non-seawater conditions. In Gemmiza 11, one band and in Misr 1, three bands were detected under seawater stress. Eustress with H_2_O_2_ provoked the appearance of one band in Gemmiza 11 and three bands in Misr 1 grown under seawater stress conditions. Eustress with H_2_O_2_ initiated again one band in both stressed wheat cultivars which were disappeared under seawater stress (Appendix A, Figure 6B).

Primer ISSR-3 indicated that under non-stressed conditions, H_2_O_2_ pretreatment promoted the appearance of three bands in Gemmiza 11 and seven bands in Misr 1. Treatment with H_2_O_2_ seawater resulted in the appearance of three bands in Gemmiza 11 and four bands in Misr 1. Under seawater stress conditions, eustress led to the synthesis of four bands in Gemmiza 11 and ten bands in Misr 1. Eustress with H_2_O_2_ resulted in the reappearance of six bands in Gemmiza 11 and six bands in Misr 1 which disappeared under seawater stress (Appendix A, Figure 6C).

Primer ISSR-4 displayed that under non-saline conditions, eustress with H_2_O_2_ stimulated the synthesis of two bands in Gemmiza 11 and six bands in Misr 1. In Gemmiza 11, two bands and in Misr 1, four bands were synthesized under seawater stress. Eustress with H_2_O_2_ induced the appearance of two bands in salinized Gemmiza 11 and eight bands in salinized Misr 1. In Gemmiza 11, two bands and in Misr 1, three bands disappeared under seawater stress but appeared again when eustress salinized both wheat cultivars with H_2_O_2_ (Appendix A, Figure 6D).

The pattern of primer ISSR-5 illustrated that under control conditions, eustress with H_2_O_2_ led to the production of two bands in Gemmiza 11 and six bands in Misr 1. In Gemmiza 11, two bands and in Misr 1, four bands were detected under seawater stress. Eustress with H_2_O_2_ caused the appearance of two bands in salinized Gemmiza 11 and seven bands in salinized Misr 1. Eustress with H_2_O_2_ led to the reappearance of four bands which disappeared in salinized Misr 1 (Appendix A, Figure 6E).

The pattern of ISSR-6 demonstrated that eustress with H_2_O_2_ provoked the synthesis of two bands in both Gemmiza 11 and Misr 1 cultivars. Under seawater stress, no bands were detected in Gemmiza 11 while, in Misr 1, two bands were synthesized under the same treatment. Eustress with H_2_O_2_ resulted in the appearance of two bands in salinized Gemmiza 11 and four in salinized Misr 1. Eustress with H_2_O_2_ resulted in the reappearance of two bands which disappeared in salinized Misr 1 (Appendix A, Figure 6F).

## 3. Discussion

In high salinity, reduction in the water potential and increment of the concentration of salts in the plant growth medium show decreased root growth. Under these conditions, most of the root energy is used to absorb the active nutrients needed, resulting in reduced root growth. On the other hand, salt stress and subsequent reduction of water potential diminish the rate of elongation and cell turgor. This is the main cause of growth loss [29,30]. In addition, the results of this study show the harmful effects of seawater stress on plant height. Reduced plant height in response to seawater stress is related to a decrease in cell elongation which itself derives from the inhibitory effects of water scarcity on the growth-regulating agents that reduce cellular swelling, cell volume and ultimately cell growth [31,32]. Although adaptation to environmental stresses is considered a complex phenomenon, our results show that salinity could be harmful to plants, however, pretreatment with H_2_O_2_ could be responsible for reducing the deleterious effects of salinity on plant growth [33,34,35,36]. H_2_O_2_ provides better root carbohydrate by increasing the activity of starch hydrolyzing enzymes [37]. The increased root length usually increases the absorption of water and nutrients, so, it seems that under salinity stress conditions, plants treated with H_2_O_2_ with increasing root length could prevent the harmful effects of salinity on growth parameters [38]. On the other hand, the results of Ashraf et al. [38] demonstrated that H_2_O_2_ can act as an osmotic adjustment agent. Our results also show that H_2_O_2_ pretreatment enhances shoot length under seawater stress conditions. As mentioned above, H_2_O_2_ regulates intracellular osmotic pressure, and as a result of decreased water loss with H_2_O_2_ pretreatment, the plants showed normal growth under seawater stress and showed the least damage in growth related traits [39]. The stimulatory impact of H_2_O_2_ on improving plant height could be attributed to energizing of the cell division and formation of the secondary cell wall [39,40].

The reduction of Chl a and Chl b content under seawater conditions in the present study may be due to the chloroplastic injury and distortion in chlorophyll ultrastructures by ROS [41]. Another reason for the reduction of chlorophyll content due to salinity stress could probably be due to the change in the pathway of nitrogen metabolism to synthesize compounds such as proline, which is used for osmotic regulation [42]. Interestingly, in the present study, salinity decreased chlorophyll content, but proline increased which confirmed the above statement about the reason for the decrease in chlorophyll content. Seed eustress with H_2_O_2_ reduced the degradation of chloroplast membrane and thus prevented the reduction of chlorophyll content under seawater stress by reducing the oxidative stress and increasing the antioxidant capacity of the cell, thereby further preventing the chlorophyll catabolism [33]. Carotenoids, fat-soluble non-enzymatic antioxidants that support the cell against free radicals and singlet oxygen, are decreased under seawater stress in both wheat cultivars and this decrease in Misr 1 was far fewer than in Gemmiza 11. Minguez-Mosquera et al. [43] stated that the reduction of carotenoids under salt stress conditions could be due to the beta-carotene degradation and formation of the zeaxanthin in the xanthophyll cycle. Ziaf et al. [44] also stated that the level of carotenoids showed a positive correlation with salt stress tolerance and has been introduced as a salt tolerance assessing index. Seed eustress with H_2_O_2_ caused a significant increase in carotenoids of two salinized wheat cultivars which might be an indication of non-enzymatic antioxidant defense.

Seawater stress increased the proline content in both wheat cultivars especially in Misr 1. In accordance with the above results, Forlani et al. [45] reported increased proline under osmotic stress conditions. The reason behind this could be that increasing the amount of proline under salinity stress was related to its osmotic and antioxidant properties under stress conditions. In the present study, eustress with H_2_O_2_ increased proline in salinized Misr 1. He and Gao [46] reported that proline rapidly increased by pretreatment of wheat seeds with H_2_O_2_. Proline accumulation through the H_2_O_2_ pretreatment in Misr 1 cultivar could be due to its efficiency in neutralizing free radicals of hydroxyl. Interestingly, H_2_O_2_ eustress retarded the accumulation of proline in salinized Gemmiza 11. Accordingly, proline emerged as a sensor of salt tolerance in Misr 1 and a symptom of the salt stress injury in Gemmiza 11.

In this study, salinity increased Na^+^ and decreased other essential nutrients significantly. Different studies showed that under seawater stress conditions, high Na^+^ and Cl^−^ absorption competes with the K^+^, Mg^2+^ and Ca^2+^ uptake which leads to a deficiency of these ions and an imbalance [16,47,48], which is in line with this study. In the present study, eustress of H_2_O_2_ not only allayed the harmful effect of excessive Na^+^ by limiting its uptake but also triggered a significant increase in the uptake of essential mineral elements including K^+^, Ca^2+^ and Mg^2+^. These findings clearly show that H_2_O_2_ activates changes, primarily linked to the stimulation of antioxidants, stands fast in safeguarding turgor and meets plant nutritional demands to thrive under salinity [49].

Under seawater stress, the level of MDA was observed as increased in both wheat cultivars indicating cell membrane damage in both cultivars, however, the accumulation of MDA in Misr 1 was lower compared to that in Gemmiza 11. These results have indications that Misr 1 possessed better protection against oxidative damage caused by seawater treatment and lower lipid peroxidation and the reduced membrane permeability compared to Gemmiza 11. Conversely, eustress of seeds with H_2_O_2_ showed a reduction in MDA in both tested wheat cultivars. The results here show that pretreatment of H_2_O_2_ can be helpful for plants in reducing oxidative stress. Subsequently, a significant reduction of MDA in the studied treatments may further ensure the integrity of the membrane and reduction of the leakage of important ions [49].

CAT activity increased under seawater stress in both cultivars and this increase was more obvious in Gemmiza 11 than Misr 1. However, this higher activity of CAT in Gemmiza 11 did not keep plenty guard against ROS, as assessed by simultaneous augmentation of MDA. Eustress with H_2_O_2_ increased the activity of this enzyme in both cultivars over salinized plants. Çavusoglu and Kabar [34], Gondim et al. [50] and Santhy et al. [51] have apprised that pretreatment of H_2_O_2_ increased the activity of CAT enzyme in plants in response to salt stress and enhanced salt tolerance in plants. In this study, POD activity increased under seawater stress in Misr 1 and decreased in Gemmiza 11 in the same situation. In previous studies, POD has been shown to play a role in the metabolism of ROS and the cellular biosynthesis of plants by hastening the final step of lignin synthesis [52,53]. This decrease in POD lessened the ability of Gemmiza 11 to scavage O_2_^−^ radicals and helped the accumulation of ROS, which could produce membrane injury. In opposition to seawater stress, H_2_O_2_ application stimulated POD activity and may has forecasted an augmented production of lignins and related defensive compounds in mitigating the oxidative pressure provoked damage. APX activity boosted under seawater stress in both tested wheat cultivars. Reports have manifested that APX activity augmented during oxidative stress in alfalfa and rice plants [54,55]. Pretreatment of H_2_O_2_ increased the activity of this enzyme in both cultivars over salinized plants. Shigeoka et al. [56] reported that pretreatment of H_2_O_2_ increased the activity of APX in the plants in response to salt stress and enhanced the salt tolerance in plants. In this work, greater activity of antioxidant enzymes in H_2_O_2_ plants compared with salinized plants was associated with dropped accumulation of MDA, indicating lower oxidative damage in H_2_O_2_ plants.

In order to verify the morpho-physiological and biochemical changes, molecular markers were used in this study to analyze and confirm the variation among the treatments. It has been discussed in various sources that ROS caused by salinity stress can severely destroy cellular components such as lipids, proteins and DNA [57,58]. Under salinity stress, all the main components of DNA (i.e., purine and pyrimidine bases, sugars and phosphodiester bonds), could suffer from damage [59]. Saha et al. [60] stated that DNA damage was provoked in response to salinity stress. Oxidative stress due to salinity stress could cause protein denaturation and even break DNA strands [61]. Regarding the mentioned cases, the molecular study (simultaneously with the study of morphological traits) of treatments could increase our understanding of salinity stress. The ISSR markers have been used in various salinity tolerance studies [25,62,63]. Also, various researchers indicated that this technique based on the PCR reaction is a fantastic tool [64] for genetic improvement of crop plants to tolerate various environmental stresses [65,66,67]. Also, different researches performed association analysis to find the ISSR markers which are linked with salinity tolerance [64,68,69,70].

Like previous studies, the results from this study have also shown superiority and repeatability of ISSR markers for determining salinity tolerance treatments primers. Considering the high percentage of polymorphism (80%), it could be expected that these markers may have acted as a powerful tool in identifying and distinguishing the treatments [71]. In their study, Krupa-Małkiewicz and Bienias [68] found that both ISSR and Random Amplification of Polymorphic DNA (RAPD) primers successfully detected the association with changes induced by chemical mutagenesis and salinity. Like this study, they concluded that the bulked segregant analysis (BSA) technique using an ISSR marker is a rapid tool for detecting salt-tolerant genotypes. Like our study, BSA technique is widely used for the salt tolerance traits in wheat [72], preharvest sprouting resistance traits in rye [73] and sex-related traits in jojoba [74].

According to the results in Appendix A, the tolerant cultivar (Misr 1) showed more bands than the sensitive cultivar (Gemmiza 11) under seawater stress. Our results are in harmony with El-Nahas et al. [75] who used the ISSR method to detect some molecular markers linked with drought tolerance in six local and exotic lentil genotypes. It could be interestingly to note that some DNA bands disappeared under seawater stress but were produced again by H_2_O_2_ treatment of the salinized wheat genotypes, especially Misr 1. Thus, these bands that are related to H_2_O_2_ treatment might have played a key role in the signaling of plant adaptive responses to seawater stress.

In conclusion, the inhibitory impacts of seawater stress on seedling growth and other relevant physiological metabolites can be mitigated by eustress seeds with H_2_O_2_. Plant growth induction and salt tolerance by H_2_O_2_ eustress in both wheat cultivars have a strong association with the variation in ISSR markers. To the best of our knowledge, this is the first report explaining the impact of H_2_O_2_ on ISSR markers of wheat under seawater stress and further molecular studies can stipulate information on the influence of H_2_O_2_ on plant metabolism under seawater stress.

## 4. Materials and Methods

### 4.1. Seed Selection and Treatments

Seeds of wheat cultivars, Gemmiza 11 and Misr 1, were provided by the Wheat Research Department, Field Crops Research Institute, Giza, Egypt. Gemmiza 11 was a salt-sensitive cultivar, whereas Misr 1 was a salt-tolerant cultivar [76]. Seawater was collected from the Red Sea of Hurghada coastal area of Egypt. The concentration of cations and anions of seawater was as follows (mg 100 mL^−1^): Na^+^ = 1078.4; K^+^ = 39.7; Ca^2+^ = 43; Cl^−^ = 1945.2; and SO_4_^2−^ = 272.1. Seeds of wheat cultivars were surface sterilized with 0.1% mercuric chloride (HgCl_2_) for 5 min, rinsed thrice with distilled water and then divided into two sets before the application of seawater stress and eustress treatments. For eustress, each of the obtained sets was soaked separately in distilled water and H_2_O_2_ (1 mM), respectively, for 8 h followed by air drying for 2 h. After the eustress treatment, seeds were washed before placing them in Petri dishes for testing growth parameters. For seedling germination, 30 seeds were placed in each sterilized Petri dish having filter paper moistened with 10 mL of distilled water or seawater and were incubated at 25 °C. Experiments were performed on each cultivar by dividing them into the following four different treatments: (i) distilled water-eustress + distilled water = W + W (0%), (ii) H_2_O_2_-eustress + distilled water = H_2_O_2_ + W (0%), (iii) distilled water-eustress + 35% diluted seawater = W + SW (35%), and (iv) H_2_O_2_-eustress + 35% diluted seawater = H_2_O_2_ + SW (35%). Three replicates (*n* = 3) were used for each treatment. Distilled water or seawater solutions (3 mL) were added to the respective Petri dishes on the 3rd and 6th days after the imbibition of the seeds. The eustress time, as well as the concentrations of H_2_O_2_ and seawater, were selected based on a series of preliminary experiments. Seedlings were harvested after a period of 10 days and a part of each treatment was transferred to a −80 °C freezer for DNA extraction. The length of fresh roots and shoots of these seedlings was also recorded. Dry weight (DW) of seedlings was determined after drying the freshly harvested seedlings in an aerated oven at 70 °C.

### 4.2. Determination of Carotenoids and Chlorophyll Contents

Carotenoid and chlorophyll (Chl a and Chl b) contents were determined in fresh leaves using a spectrophotometer by following the method adopted by Lichtenthaler and Wellburn [77].

### 4.3. Determination of Proline and Malondialdehyde Contents

Bates et al. [78] method was used to measure the proline contents in fresh leaves, while the thiobarbituric acid (TBA) reaction was used to determine malondialdehyde (MDA) content in the fresh leaf tissues following the method described by Abdel Latef and Tran [79]. The absorbance was read at 450, 532 and 600 nm, and the MDA content was calculated on the fresh weight (FW) basis using the following formula:MDA content (nmol g^−1^ FW) = 6.45 × (*A*_532_ − *A*_600_) − 0.56 × *A*_450_

### 4.4. Determination of Mineral Contents

Dried seedling samples (0.1 g) were acid-digested using 80% perchloric acid (HClO_4_) and concentrated using H_2_SO_4_ solution (1:5) for 12 h. The Ca^2+^, Mg^2+^, Na^+^ and K^+^ contents in the digested samples were determined as described by Williams and Twine [80].

### 4.5. Determination of Antioxidant Enzyme Activities

Fresh leaf samples were used to determine the activity of antioxidant enzymes. Extraction of samples and preparation of supernatants were carried out according to the method reported in Ahmad et al. [81]. Ascorbate peroxidase (APX; EC 1.11.1.11), peroxidase (POD; EC 1.11.1.7) and catalase (CAT; EC 1.11.1.6) activities were assessed according to the methods described by Chen and Asada [82], Maehly and Chance [83] and Aebi [84], respectively.

### 4.6. DNA Extraction and ISSR-PCR Analysis

DNA extraction and purification were carried out according to the procedure of the DNeasy Kit (Qiagen, Hilden, Germany). A set of 6 primers were used in the ISSR-PCR technique (Table 2). The polymerase chain reaction (PCR) was carried out as described in Adhikari et al. [85]. The PCR products were separated by electrophoresis using a 1% agarose gel and photographed using a Gel Documentation System (BIO-RAD 2000). Lambda DNA Hind III digest was used as a DNA marker.

### 4.7. Statistical Analysis

Analysis of variance (ANOVA) of all the traits was performed based on the factorial design using the SAS ver. 9 software [86]. The mean comparison of the treatments was investigated using Duncan’s multiple range test (DMRT) at the level of significance (*p* < 0.05) using the SPSS ver. 19 [87]. The data obtained from three replications (*n* = 3) was presented as the means ± standard errors (SEs) and different letters were used to show the significant different treatment bars. Hierarchical cluster analysis was conducted using Past Software ver. 2.12 [88]. Principal component analysis (PCA) was performed to draw a biplot of the studied treatments using StatGraphics X VII version [89].

## Figures and Tables

**Figure 1 plants-08-00303-f001:**
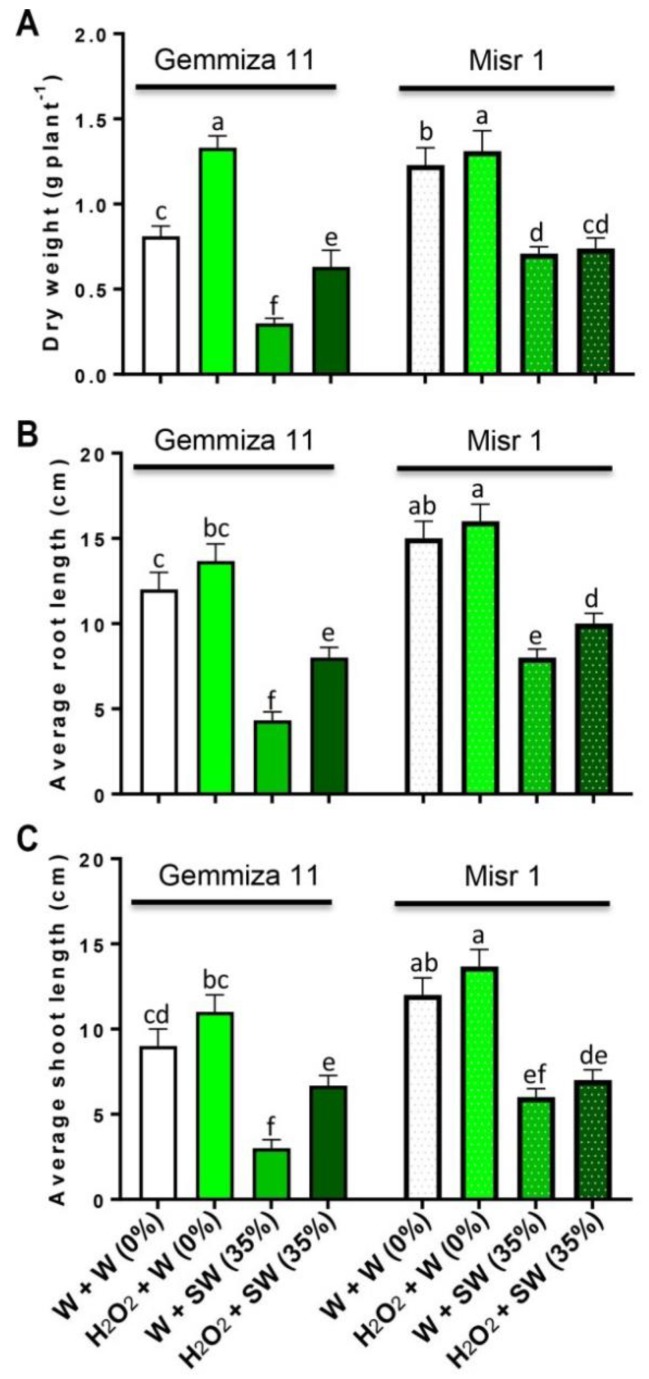
Evaluating the effects of different treatments including distilled water-eustress + distilled water = W + W (0%), H_2_O_2_-eustress + distilled water (%) = H_2_O_2_ + W (0%), distilled water-eustress + 35% seawater = W + SW (35%) and H_2_O_2_-eustress + 35% seawater = H_2_O_2_ + SW (35%) on (**A**) dry weight (DW), (**B**) root length and (**C**) shoot length of two different wheat cultivars, Gemmiza 11 and Misr 1. Bars represent means of three (*n* = 3) replicates with standard errors (SEs) and different letters on the bars indicate statistically significant difference following Duncan’s multiple range test at the level of significance (*p* < 0.05).

**Figure 2 plants-08-00303-f002:**
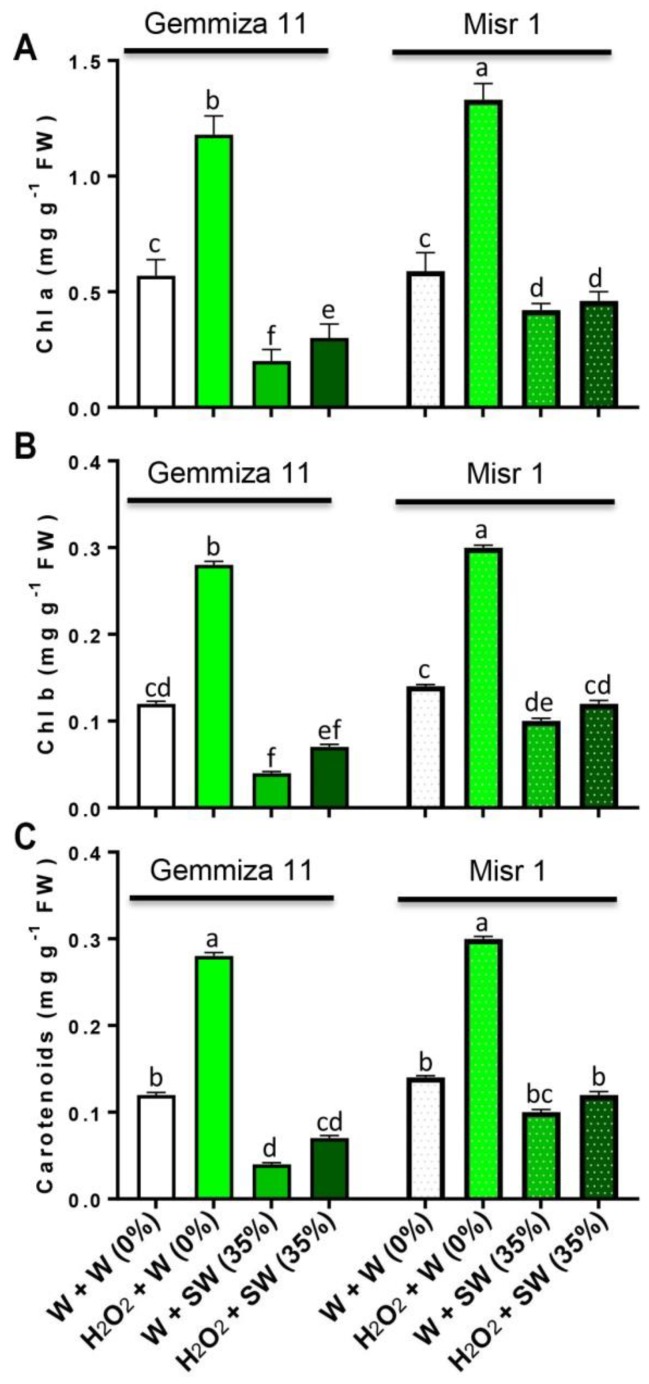
Evaluating the effects of different treatments including distilled water-eustress + distilled water = W + W (0%), H_2_O_2_-eustress + distilled water (%) = H_2_O_2_ + W (0%), distilled water-eustress + 35% seawater = W + SW (35%) and H_2_O_2_-eustress + 35% seawater = H_2_O_2_ + SW (35%) on (**A**) chlorophyll (Chl) a, (**B**) chlorophyll b and (**C**) carotenoids of two different wheat cultivars, Gemmiza 11 and Misr 1. Bars represent means of three (*n* = 3) replicates with standard errors (SEs) and different letters on the bars indicate statistically significant difference following Duncan’s multiple range test at the level of significance (*p* < 0.05).

**Figure 3 plants-08-00303-f003:**
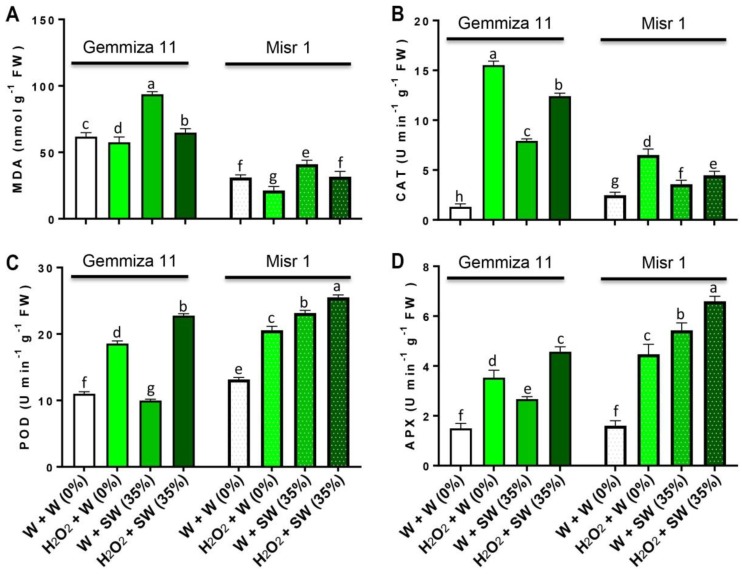
Evaluating the effects of different treatments including distilled water-eustress + distilled water = W + W (0%), H_2_O_2_-eustress + distilled water (%) = H_2_O_2_ + W (0%), distilled water-eustress + 35% seawater = W + SW (35%) and H_2_O_2_-eustress + 35% seawater = H_2_O_2_ + SW (35%) on (**A**) malondialdehyde (MDA) content, (**B**) catalase (CAT) activity, (**C**) peroxidase (POD) activity and (**D**) ascorbate peroxidase (APX) activity of two different wheat cultivars, Gemmiza 11 and Misr 1. Bars represent means of three (*n* = 3) replicates with standard errors (SEs) and different letters on the bars indicate statistically significant difference following Duncan’s multiple range test at the level of significance (*p* < 0.05).

**Figure 4 plants-08-00303-f004:**
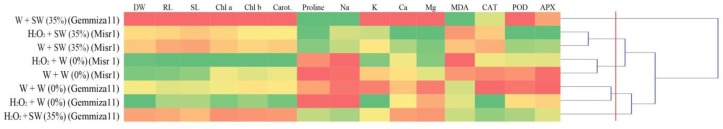
Hierarchical clustering to understand treatment-variable relationships of two wheat cultivars, Gemmiza 11 and Misr 1, under different treatment combinations including distilled water-eustress + distilled water = W + W (0%), H_2_O_2_-eustress + distilled water (%) = H_2_O_2_ + W (0%), distilled water-eustress + 35% seawater = W + SW (35%) and H_2_O_2_-eustress + 35% seawater = H_2_O_2_ + SW (35%). Abbreviations are as follows: DW = dry weight, RL = root length, SL = shoot length, Chl a = chlorophyll a, Chl b = chlorophyll b, Carot = carotenoids, MDA = malondialdehyde, CAT = catalase, POD = peroxidase, APX = ascorbate peroxidase.

**Figure 5 plants-08-00303-f005:**
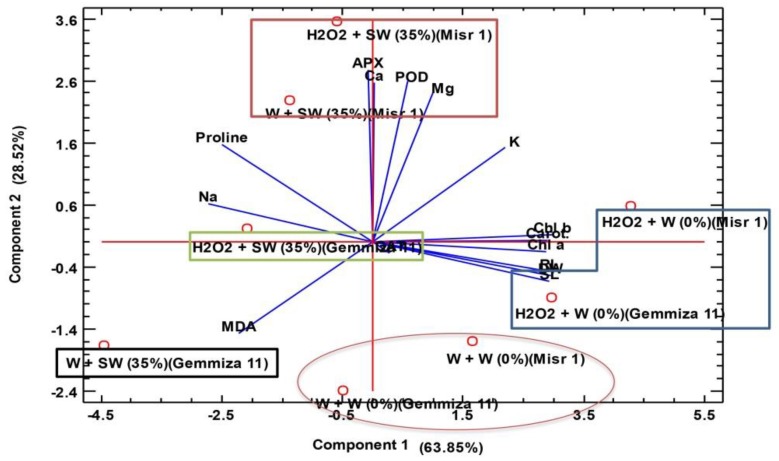
Principle component analysis (PCA) to understand treatment–variable relationships of two wheat cultivars, Gemmiza 11 and Misr 1, under different treatment combinations including distilled water-eustress + distilled water = W + W (0%), H_2_O_2_-eustress + distilled water (%) = H_2_O_2_ + W (0%), distilled water-eustress + 35% seawater = W + SW (35%) and H_2_O_2_-eustress + 35% seawater = H_2_O_2_ + SW (35%). Abbreviations are as follows: DW = dry weight, RL = root length, SL = shoot length, Chl a = chlorophyll a, Chl b = chlorophyll b, Carot = carotenoids, MDA = malondialdehyde, CAT = catalase, POD = peroxidase, APX = ascorbate peroxidase.

**Figure 6 plants-08-00303-f006:**
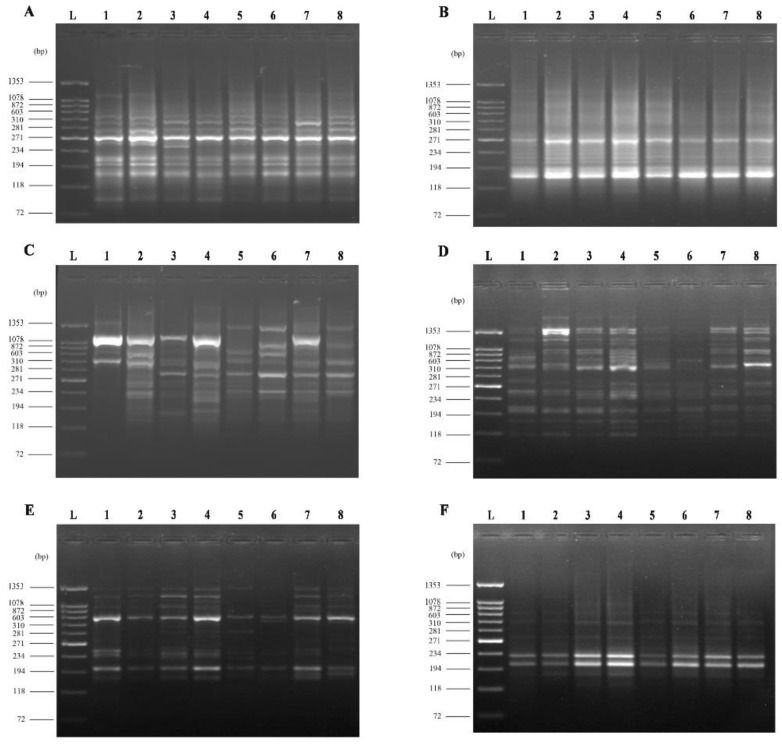
Fingerprint of the studied treatments using inter-simple sequence repeat (ISSR) markers. (**A**) ISSR-1; (**B**) ISSR-2; (**C**) ISSR-3; (**D**) ISSR-4; (**E**) ISSR-5; (**F**) ISSR-6; L—ladder; 1—W + W (0%)—Gemmiza 11; 2—H_2_O_2_ + W (0%)—Gemmiza 11; 3—W + SW (35%)—Gemmiza 11; 4—H_2_O_2_ + SW (35%)—Gemmiza 11; 5—W + W (0%)—Misr 1; 6—H_2_O_2_ + W (0%)—Misr 1; 7—W + SW (35%)—Misr 1; 8—H_2_O_2_ + SW (35%)—Misr 1.

**Table 1 plants-08-00303-t001:** Mean comparison of the proline (mg g^−1^ FW) and nutrients (mg g^−1^ DW) under different treatments.

Treatments	Proline	Na^+^	K^+^	Ca^2+^	Mg^2+^
W + W (0%) (Gemmiza 11)	0.50 ^d^ ± 0.05	2.92 ^d^ ± 0.44	8.50 ^d^ ± 1.51	0.93 ^d,e^ ± 0.05	0.23 ^e,f^ ± 0.03
H_2_O_2_ + W (0%) (Gemmiza 11)	0.17 ^f^ ± 0.02	2.23 ^d^ ± 0.11	17.73 ^a^ ± 2.26	1.16 ^c,d^ ± 0.06	0.36 ^d^ ± 0.02
W + SW (35%) (Gemmiza 11)	1.80 ^a^ ± 0.03	36.64 ^a^ ± 3.30	7.40 ^d^ ± 1.05	0.53 ^f^ ± 0.02	0.17 ^f^ ± 0.02
H_2_O_2_ + SW (35%) (Gemmiza 11)	1.25 ^c^ ± 0.05	24.78 ^b^ ± 4.60	14.00 ^b,c^ ± 4.29	0.78 ^e,f^ ± 0.05	0.30 ^d,e^ ± 0.05
W + W (0%) (Misr 1)	0.17 ^f^ ± 0.04	2.63 ^d^ ± 0.08	12.37 ^c^ ± 0.87	1.06 ^d^ ± 0.03	0.71 ^c^ ± 0.06
H_2_O_2_ + W (0%) (Misr 1)	0.38 ^e^ ± 0.07	2.31 ^d^ ± 0.09	17.00 ^a,b^ ± 1.51	1.41 ^c^ ± 0.03	1.24 ^b^ ± 0.05
W + SW (35%) (Misr 1)	1.59 ^b^ ± 0.07	20.89 ^c^ ± 2.05	13.63 ^b,c^ ± 0.25	2.89 ^b^ ± 0.40	1.28 ^b^ ± 0.03
H_2_O_2_ + SW (35%) (Misr 1)	1.76 ^a^ ± 0.03	17.48 ^c^ ± 0.89	15.20 ^a–c^ ± 0.96	4.53 ^a^ ± 0.16	1.45 ^a^ ± 0.10

Water-eustress + distilled water = W + W (0%), H_2_O_2_-eustress + distilled water (%) = H_2_O_2_ + W (0%), distilled water-eustress + 35% seawater = W + SW (35%) and H_2_O_2_-eustress + 35% seawater = H_2_O_2_ + SW (35%). The data represent means of three (*n* = 3) replicates with standard errors (SEs) and different letters on the data indicate statistically significant difference following Duncan’s multiple range test at the level of significance (*p* < 0.05). DW: dry weight.

**Table 2 plants-08-00303-t002:** Representation and sequence of inter-simple sequence repeat (ISSR) primers.

Primer Name	Sequence	Motif	OB	PB	P%
ISSR-1	5’-AGAGAGAGAGAGAGAGC-3’	(AG)_8_C	18	13	72.22
ISSR-2	5’-ACACACACACACACACT-3’	(AC)_8_T	11	6	54.54
ISSR-3	5’-ACACACACACACACACG-3’	(AC)_8_G	25	23	92
ISSR-4	5’-CGCGATAGATAGATAGAT-3’	CGC(GATA)_4_	17	14	82.35
ISSR-5	5’-GACGATAGATAGATAGATA-3’	GAC(GATA)_4_	19	16	84.21
ISSR-6	5’-GACAGACAGACAGACAAT-3’	(GACA)_4_AT	11	9	81.81

OB: number of observed bands; PB: number of polymorphic bands; P%: polymorphism percentage.

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
