# Peer review of "Eustress with H2O2 Facilitates Plant Growth by Improving Tolerance to Salt Stress in Two Wheat Cultivars"

_plants, 2019, doi:10.3390/plants8090303_

Round 1

Reviewer 1 Report

Still the discussion is poor.

Table 2 is too long. Present it as other kinds of illustration.

Author Response

POINT 1 Still the discussion is poor.

Response: The discussion is improved by new references.

POINT 2 Table 2 is too long. Present it as other kinds of illustration.

Response: The Table is moved to Supplementary Files.

Reviewer 2 Report

REVIEWER 2

Comment 1

The article address a very interesting topic about how priming with H2O2 improves tolerance to salinity stress in wheat. Nevertheless, most of the biochemical measurements have already been performed in other cereal or even other Wheat cultivars, reaching similar results. Therefore, my major concern about this article is the lack of originality for the quality standard of Plants.

Response

We would like to give you a special thanks for your careful reading of the whole MS and also for your meaningful corrections that helped us a lot to improve our manuscript significantly. In this study, seed priming with H2O2 in mitigating the adverse impacts of seawater stress has been evaluated in two wheat (Triticum aestivum L.) cultivars, namely Gemmiza 11 as a salt-sensitive and Misr 1 as a salt-tolerant cultivar, with contrasting phenotypes in response to the seawater stress. In the other studies, only NaCl stress have been applied on the plants, but in this study we have applied seawater (from the Red Sea of Hurghada coastal area of Egypt containing Na+ = 1078.4; K+ = 39.7; Ca2+ = 43; Cl- = 1945.2 and SO42- = 272.1 mg/lit) on these wheat cultivars. Furthermore, the experiments were performed at molecular level using six ISSR markers to observe the differential responses of the two wheat cultivars against seawater stress with or without H2O2 -priming. In fact, it can be said that this is the first report for the application of ISSR markers in determining the effect of H2O2 -priming on an individual at the molecular level. Results from these experiments helped us in understand the impact of H2O2 priming on reducing the adverse effects of seawater stress on tested two wheat cultivars and other cereal crops as well as different wheat cultivars.

Reviewer 2

I realized that the experiments were done using seawater instead of NaCl solutions, nevertheless, the effect is exactly the same as NaCl solutions used by others authors in so many other cultivars and wheat, probably due to the content of Na+ and Cl- is similar. That is my major concern and why I can not see any new or innovative results that have not been described previously.

As I told, the molecular studies with ISSR are the most important part, they show the H2O2 treatments and seawater may play genetic variability in these cultivars, but much further research is needed. Is DNA damaged? Which are the specific genes involved?

Comment 2

Some data from Figure 1 and 2 are not statistically significant, while authors still affirm they are. Eg. Seedlings DW, Chl a, Chl b, shoots length in Misr 1 cultivar showed no significant differences between Seawater and Seawater+ H2O2. I would also recommend expressing the graphs in %, (or the text in the same measures as the figures), since for the reader, it is hard to follow the measurements given in the text.

Response

The dear reviewer is correct. But in this experiment, we have compared the two cultivars with each other and the mean value for them were significant.

Reviewer 2

As far as I see in the text and the graphs, you are comparing treatments in both cultivars, no between cultivars. All represented and explained “in comparison with the respective control plants”. for example, Misr 1 does not show significant differences between w+sw and priming H2O2 +SW in DW, since they indicate d and cd  on the bars.  Which is the 4% enhancement you explain in the text. That´s not significant different. And similar happen in Chlb, Carotenoids, Chla, shoots, etc.

And it is very confusing the percentage data in the text, while the graphs are in units. Graphs should be improved in that sense.

Comment 3

The most interesting part of the article is the use of ISSR analysis to check out the polymorphic bands within the four treatments. Nevertheless, I wonder which are those loci that appears or disappears with H2O2 priming and seawater. Thus, a genetic linkage map of these cultivars, using quantitative trait loci (QTLs) and these ISSR markers, would better serve the scientific community and the article would gain considerably in terms of originality.

Response

The dear reviewer is correct. But for QTL mapping, we need structured population or a huge number of cultivars. But in this study, we have only 2 cultivars and different treatments. On the 2 other hand, we need a saturated linkage map (using a lot of markers) to map the traits carefully. Also, some additional techniques and statistical analysis must be done to evaluate genotypes for salt tolerance, so we can focus this point in next experiments using data in this manuscript as a starting point.

Reviewer 2

Although ISSR results are promising data, those experiments would do this manuscript worthy and would improve its originality with innovative results.

Author Response

Response to Reviewer 2 Comments

POINT 1 I realized that the experiments were done using seawater instead of NaCl solutions, nevertheless, the effect is exactly the same as NaCl solutions used by others authors in so many other cultivars and wheat, probably due to the content of Na+ and Cl- is similar. That is my major concern and why I can not see any new or innovative results that have not been described previously.

As I told, the molecular studies with ISSR are the most important part, they show the H2O2 treatments and seawater may play genetic variability in these cultivars, but much further research is needed. Is DNA damaged? Which are the specific genes involved?

Response: That’s correct. We will consider it in future studies.

POINT 2 As far as I see in the text and the graphs, you are comparing treatments in both cultivars, no between cultivars. All represented and explained “in comparison with the respective control plants”. for example, Misr 1 does not show significant differences between w+sw and priming H2O2 +SW in DW, since they indicate d and cd  on the bars.  Which is the 4% enhancement you explain in the text. That´s not significant different. And similar happen in Chlb, Carotenoids, Chla, shoots, etc.

And it is very confusing the percentage data in the text, while the graphs are in units. Graphs should be improved in that sense.

Response: The data converted from percentage to the real amounts.

Reviewer 3 Report

The fact that many studies can be cited on the subject is clearly not a proof of originality, on the contrary.

I can understand that the Authors contested the term “outdated”, but they continue to use “priming”, this term should not be used anymore since 2010. Eustress should be used instead (see Kranner et al. 2010 New Phytol 10.1111/j.1469-8137.2010.03461.x).

The answers provided did not satisfied me.

Author Response

Response to Reviewer 3 Comments

POINT 1 The fact that many studies can be cited on the subject is clearly not a proof of originality, on the contrary.

I can understand that the Authors contested the term “outdated”, but they continue to use “priming”, this term should not be used anymore since 2010. Eustress should be used instead (see Kranner et al. 2010 New Phytol 10.1111/j.1469-8137.2010.03461.x).

The answers provided did not satisfied me.

Response: Thanks for your comments. The manuscript is improved by new references for discussion part. Also the title and text is changed based on your advice.

Round 2

Reviewer 2 Report

I'm sorry, but my major concern is the lack of originality, and the article resubmitted has not been improved in that regard. Just minor revision was done.

Author Response

Response to Academic Editor Notes

POINT 1 - the discussion is very long and would benefit from being much more concise

Response: We have trimmed the discussion in revised version.

POINT 2- the part of the results on the molecular analysis is very long, many of the details written out in the text on number of polymorphic bands, band sizes etc, could be summarised in a table

Response: We removed all number of polymorphic bands, band sizes etc from text and showed that in updated Table 2 and Supplementary File 1.

Reviewer 3 Report

The Authors have improved the quality of the discussion of the present manuscript.

This manuscript is, in my opinion, now acceptable for publication.

Author Response

Response to Academic Editor Notes

POINT 1 - the discussion is very long and would benefit from being much more concise

Response: We have trimmed the discussion in revised version.

POINT 2- the part of the results on the molecular analysis is very long, many of the details written out in the text on number of polymorphic bands, band sizes etc, could be summarised in a table

Response: We removed all number of polymorphic bands, band sizes etc from text and showed that in updated Table 2 and Supplementary File 1.

This manuscript is a resubmission of an earlier submission. The following is a list of the peer review reports and author responses from that submission.

Round 1

Reviewer 1 Report

In my opinion the present study lacks of originality and technical difficulties. Although the problem of priming and resistance to stress including salt stress are hot research topics, the experimental approaches used are outdated. The use of ISSRs is outdated and known to lack reproducibility. A clear justification for the choice of these markers have to be provided. Therefore the Authors have to carefully revise their work, and clearly highlight and discuss the originality of their work, how their work will contribute to the field and why it should be publish. The conclusion have to be revised accordingly.  

Reviewer 2 Report

The article address a very interesting topic about how priming with H2O2 improves tolerance to salinity stress in wheat. Nevertheless, most of the biochemical measurements have already been performed in other cereal or even other Wheat cultivars, reaching similar results. Therefore, my major concern about this article is the lack of originality for the quality standard of Plants.

Some data from Figure 1 and 2 are not statistically significant, while authors still affirm they are. Eg. Seedlings DW, Chl a, Chl b, shoots length in Misr 1 cultivar showed no significant differences between Seawater and Seawater+H2O2. I would also recommend expressing the graphs in %, (or the text in the same measures as the figures), since for the reader, it is hard to follow the measurements given in the text.

The most interesting part of the article is the use of ISSR analysis to check out the polymorphic bands within the four treatments. Nevertheless, I wonder which are those loci that appears or disappears with H2O2 priming and seawater. Thus, a genetic linkage map of these cultivars, using quantitative trait loci (QTLs) and these ISSR markers, would better serve the scientific community and the article would gain considerably in terms of originality.

Reviewer 3 Report

This manuscript presents some interesting results on the effect of h2O2 priming in conferring slt stress tolerance.

There are plenty of research on H2O2 priming but in this paper there are molecular aspects and they adds some extra values.

However, this paper needs drastic revisions before further processing:

 In the title salt stress is  enough because seawater is ultimately NaCl.

Use two wheat cultivars instead of wheat cultivars.

Figures are too small and not understandable. Please split them

Although there are some significant differences but CAT, MDA reduction etc are not so improved. What is the main effect of H2O2 priming?

Results should be concise.

Discussion needs improvement. Many of the recent results are overlooked e.g. Sayed, S. & Gadallah, M. Acta Physiol Plant (2019) 41: 113. https://doi.org/10.1007/s11738-019-2901-2

Hasanuzzaman et al. (2017) Hydrogen Peroxide Pretreatment Mitigates Cadmium-Induced Oxidative Stress in Brassica napus L.: An Intrinsic Study on Antioxidant Defense and Glyoxalase Systems. Front. Plant Sci. 8:115. doi: 10.3389/fpls.2017.00115

7. ISSR analysis can be added as supplementary file.